# Evaluation of leaf-level optical properties employed in land surface models

Titta Majasalmi[1], Ryan M. Bright[1]

[1]Norwegian Institute of Bioeconomy Research (NIBIO), Box 115, 1433 Ås, Norway

*Correspondence to*: Titta Majasalmi (titta.majasalmi@nibio.no)

## Abstract

Vegetation optical properties have a direct impact on canopy absorption and scattering and are thus needed for modeling surface fluxes. Although Plant Functional Type (PFT) classification varies between different land surface models (LSMs), their optical properties must be specified. The aim of this study is to revisit the 'time-invariant optical properties table' of the

Simple Biosphere (SiB) model (later referred as 'SiB-table') presented 30-years ago by Dorman and Sellers (1989) which has since become adopted by many LSMs. This revisit was needed as much of the data underlying the SiB-table was not formally reviewed or published, or was based on older papers or on personal communications (i.e. the validity of the optical property source data cannot be inspected due to missing data sources, outdated citation practices, and varied estimation methods). As many of today's LSMs (e.g. Community Land Model (CLM), Jena Scheme of Atmosphere Biosphere Coupling in Hamburg

(JSBACH), and Joint UK Land Environment Simulator (JULES)) either rely on the optical properties of the SiB-table or lack references altogether for those they do employ, there is a clear need to assess (and confirm or correct) the appropriateness of those being used in today's LSMs. Here, we use various spectral databases to synthesize and harmonize the key optical property information of PFT classification shared by many leading LSMs. For forests, such classifications typically differentiate PFTs by broad geo-climatic zones (i.e. tropical, boreal, temperate) and phenology (i.e. deciduous vs. evergreen). For short-statured

vegetation, such classifications typically differentiate between crops and grasses and by photosynthetic pathway. Using the PFT classification of the CLM (version 5) as an example, we found the optical properties of the visible band (VIS; 400-700 nm) to fall within the range of measured values. However, in the near-infrared and shortwave infrared bands (NIR+SWIR; e.g. 701-2500 nm, referred as 'NIR') notable differences between CLM default and measured values were observed, thus suggesting that NIR optical properties are in need of an update. For example, for conifer PFTs, the measured mean needle

single scattering albedo (*SSA*, i.e. sum of reflectance and transmittance) estimates in NIR were 62% and 78% larger than the CLM default parameters, and for PFTs with flat-leaves, the measured mean leaf *SSA* values in NIR were 20%, 14% and 19% larger than the CLM defaults. We also found that while the CLM5 PFT-dependent leaf angle values were sufficient for forested PFTs and grasses, for crop PFTs the default parameterization appeared too vertically oriented, thus warranting an update. In addition, we propose using separate bark reflectance values for conifer and deciduous PFTs, and demonstrate how shoot-level

clumping correction can be incorporated into LSMs to mitigate violations of turbid media assumption and Beer's law caused by non-randomness of finite-sized foliage elements.

# 1 Introduction

Vegetation optical properties have a direct impact on canopy absorption and scattering and are thus needed for modeling surface fluxes. All land surface models (LSMs) have modules to simulate radiation transfer (later referred as 'RT') of surfaces. Although there are many types of canopy RT models with varying complexities - from light extinction algorithms to those applying turbid medium and geometric-optical methods - they must specify the following: optical properties (i.e. reflectance '$R$' and transmittance '$T$') of canopy elements such as foliage and bark, canopy foliage area (e.g. Leaf Area Index ($LAI$, $m^2/m^2$)), and vegetation spatial ordering (e.g. Leaf Inclination Angle, $LIA$ i.e., angle between the leaf surface normal and the zenith). At present, most LSMs are limited to one-dimensional (1D) radiative exchange relying on solutions derived from two-stream approximations based on plane-parallel turbid media assumptions (Loew et al., 2014; Yuan et al., 2017).

30 years ago, Dorman and Sellers (1989) presented a 'time-invariant optical properties table' for the Simple Biosphere (SiB) model (later referred as 'SiB-table' or 'SiB-classes') which was compiled using available data and field notes of the time. To the best of our knowledge, some of this data, however, was either never subjected to formal peer-review and published (e.g. Miller, 1972; Klink and Willmot, 1985) or was based on earlier research citing even older papers or personal communications (i.e. the validity of the source data cannot be examined due to a lack of transparency). As many of today's LSMs (e.g. Community Land Model (CLM) (Bonan et al., 2002), land surface model developed at the Institut d'Astronomie et de Geˊophysique Georges Lemaıˆtre (IAGL) (de Ridder, 1997), Jena Scheme of Atmosphere Biosphere Coupling in Hamburg (JSBACH, 2019), and Joint UK Land Environment Simulator (JULES) (Clark et al., 2011)) either rely on the original SiB-table optical properties or on undocumented data - there is a clear need to assess (and confirm or correct) the appropriateness of PFT-dependent optical properties by benchmarking to data collected and stored using present-day research norms and documentation standards.

Measurements of $R$ and $T$ spectra ($\lambda$) of leaves and needles can be achieved using integrating spheres (e.g. Hovi et al., 2017; Lukeš et al., 2013), $R_{(\lambda)}$ of bark and short  vegetation (i.e. grasses and crops) using handheld spectrometers (e.g. Lang et al., 2002), and $LIA$ using inclinometers or digital photography (e.g. Ryu et al., 2010). Measured $R_{(\lambda)}$ and $T_{(\lambda)}$ can be averaged over different wavelength bands (e.g. visible (VIS), 400-700 nm; near-infrared and shortwave infrared (NIR+SWIR, later referred as 'NIR'), 701-2500 nm) required by LSMs, or resampled to correspond with different satellite sensor's band definitions (Asner et al., 1998). Although laboratory measurements of leaf optical properties have been done since the 1960's (Gates et al., 1965), compiling the spectra into public databases with measured other traits and metadata started relatively recently. Today's spectral libraries, such as EcoSIS (EcoSIS, 2017) and SPECCHIO (Hueni et al., 2009), are open databases for storing spectral data from different field campaigns to promote data usage by researchers and model developers. Some reputable example datasets stored in EcoSIS are 'Lopex93' (Hosgood et al., 1993) and 'Angers'(Jacquemound et al., 2003): Lopex93 and Angers contain data for species with flat leaves. For needleleaved species, spectral data is available from SPECCHIO.

Reflectance spectra of different types of (hemi-)boreal grass species communities and tree bark are available e.g. from Estonian research database by Lang et al. (2002). Although much spectral data exists and are freely available, the earlier spectral datasets suffer from not being available online (e.g. data for 26 species from herbs to trees measured in Mississippi and Kansas, USA (Knapp and Carter, 1998) or having limited wavelength range (e.g. BOREAS, comprised North American tree species, was

limited to the wavelength range of 400–1100 nm (Middleton et al., 1997)).

Similar with the developments of spectral databases, a wealth of information surrounding forest foliage *LIA* (°) has become available in recent years owed to new measurement techniques (e.g. Ryu et al., 2010). *LIA* is needed to obtain the direct beam extinction coefficient, and e.g. to separate foliage area into sunlit and shaded parts as foliage responses to diffuse and direct

solar radiation differ (Gu et al., 2002), and for RT model inversion (Combal et al., 2003). While measuring *LIA* of grasses and crops is relatively straightforward and has been conducted since 1960 using inclined point quadrats by measuring the number of vegetation contacts from which the *LIA* is estimated (Warren Wilson, 1960), methods for measuring tree foliage *LIA* have been lacking due to problems applying them to tall forest canopies (i.e. the high cost of measurements and inability to reproduce them). At present, a state-of-the-art method for determining *LIA*s is based on digital photography which allows robust, non-

destructive measurements (i.e. reproducible data) with low cost. In the absence of measured data, estimates regarding leaf angle distributions have often been obtained using modeling or assumed spherical (e.g. Oker-Blom and Kellomäki, 1982; Goudriaan, 1988). Based on compilation of measured and published data, we assess the appropriateness of the PFT-dependent *LIA* parameterization used by today's LSMs.

In recent years, LSMs have been adapted to incorporate new important processes such as nutrient cycling and land cover dynamics, while the developments in biogeophysical processes like surface radiation schemes have not developed much further (Loew et al., 2014). Criticisms have dealt with incompatibilities in vegetation structural descriptions in the employed RT schemes (e.g. MODIS LAI is based on three-dimensional (3D) RT model whereas CLM employs 1D RT model) (Loew et al., 2014), which may lead to erroneous assessments of the absorbed, transmitted and reflected fluxes (Pinty et al., 2004; 2006)).

This incompatibility can be avoided using effective state variables (i.e. effective LAI and effective optical properties), which translate the 3D vegetation information into 1D properties, and correctly represent the effects of vegetation structural heterogeneity within a grid cell (e.g. Pinty et al., 2004; Wang et al., 2018). Effective state variables can be obtained by applying corrections that take into account vegetation non-randomness (e.g. structure) or by measuring the clumped targets (e.g. conifer forest canopy (Majasalmi et al., 2017)). The problem associated with clumping is caused by the turbid media assumption and

Beer's law, which assume foliage elements to be infinitely small and randomly located - neither of which is true for non-gases. While clumping effects may appear at many scales (e.g. shoot, crown, tree, landscape) and may be corrected using various techniques (e.g. Norman and Jarvis, 1975; Chen and Black,1992; Stenberg, 1996; Smolander and Stenberg, 2003; Haverd et al., 2012; He et al., 2012; Wang et al., 2018), there is consensus regarding the existence and significance of clumping on influencing RT of a vegetation media. Noteworthy is that currently clumping effects are not accounted in LSMs.

The MODIS LAI algorithm (Knyazikhin et al., 1999), which is used to parameterize many LSMs, is based on stochastic radiative transfer equation and theory of spectral invariants, which packs 3D information into a 1D equation. This is possible as interactions between photons and canopy elements converge to invariant patterns, which can be quantified using a few wavelength independent parameters, which satisfy the law of energy conservation (Wang et al., 2018). MODIS LAI algorithm for needle forests (section 2.2.6 Biome 6 in Knyazikhin et al., 1999) assumes needles to be clustered into shoots, withshoots being further clustered into crowns. Both clumping corrections are based on spectral invariants theory, which can be interpreted as 'photon recollision probability' ($p$) (Smolander and Stenberg, 2005). Interpretation of $p$ provides physical intuition with the mathematical concept and association with measurable structural vegetation properties (e.g. Lewis and Disney, 2007; Rautiainen and Stenberg, 2005; Smolander and Stenberg, 2005). The $p$ is a probability by which a photon scattered (reflected or transmitted) from a leaf or needle in the canopy will interact within the canopy again - In a canopy composed of leaves, a photon scattered from a leaf will not re-interact with the same leaf; however, in a canopy composed of shoots, a photon scattered out from a shoot may have interacted with the needles forming the shoot multiple times. The violations of turbid media assumption and Beer's law by needles clustering into shoots, can be mitigated by changing the basic unit from a needle to a shoot (Nilson and Ross, 1997), by upscaling needle single scattering albedo spectra ($SSA_{needle(\lambda)}$) into shoot single scattering albedo spectra ($SSA_{shoot(\lambda)}$) spectra based on shoot geometry (Rautiainen et al., 2012), and by simply replacing $SSA_{needle}$ with effective $SSA_{shoot}$ in the RT calculation. This correction is applicable to models employing turbid media assumption and Beer's law, and provides simplicity required by LSMs. In addition to MODIS LAI algorithm, the $p$ is currently incorporated into different types of RT modeling schemes such as PARAS models (Stenberg et al., 2016), and Forest Reflectance and Transmittance (FRT) model (Kuusk and Nilson, 2000).

There is large variation in the way optical properties can be defined (e.g. species composition) and measured (e.g. measuring device and its wavelength range). Therefore, the main objective of this study is not to provide 'final truth' regarding PFTs optical properties; rather, our aim is to assess their appropriateness by benchmarking to data collected and stored using present-day research norms, reviewed and synthesized here. Specifically, our objectives are to: 1) verify the PFT-dependent optical properties used in today's LSMs using the CLM PFT classification and optical property look-up table as an example, 2) suggest an alternative to account for shoot-level clumping of conifers in models employing turbid media assumption and Beer's law; and 3) assess the appropriateness of the $LIA$ specification included in the CLM's (e.g. v5) optical properties table. Four supplementary files are provided to inspect the observed variation, and to recalculate the PFT-dependent means following different PFT definitions: Our recommendation for enhancing CLM5 optical properties table ('S1_CLM5.pdf'), and two source files ('S2_OP.csv' and 'S3_LIA.csv'), which contain species mean optical property (i.e. $T$, $R$ and $SSA$, and for conifers $SSA_{shoot}$) values over the VIS and NIR bands, and species mean $LIA$s (in degrees and departure from spherical + classic leaf angle type) along with references to raw data. In addition, a pdf copy of the leaf angle tables presented by Ross (1981) ('S4_Ross_1981.pdf') is included as it contains the data used in this study and references to original works.

**2 Materials and Methods**

**2.1 Pedigree of the CLM -table**

The following briefly describes the composition of the optical properties table used by today's CLM versions that is used as an example PFT-classification in this paper. The SiB-table by Dorman and Sellers (1989) was partly reused by (Bonan et al.,
2002) to suit the needs of the CLM (**Table 1.**). Bonan et al. (2002) assigned properties of SiB-table class 1 'broadleaved-evergreen trees' ('BET') and SiB-table class 2, 'broadleaved-deciduous trees' (BDT), for CLM 'BET tropical', 'BET -temperate', 'BDT temperate', 'BDT boreal', 'BDT tropical', and for PFTs containing 'broadleaved-deciduous shrubs' (BDS) (i.e. 'BDS temperate' and 'BDS boreal'). The leaf angle specification (as departure from spherical, $\chi_L$, i.e. 1= planophile, -1= erectophile, and 0= spherical) for both BET PFTs was set to 0.10, and for temperate and boreal BDTs and BDSs to 0.25.
However, for 'BDT tropical', the leaf angle was set 0.01. The SiB-class 4 'Needleleaf-evergreen trees' (NET) and class 5 'Needleleaf-deciduous trees' (NDT) were used to form CLM PFTs 'NET temperate' and 'NET boreal,' 'NDT boreal' and 'broadleaf-evergreen shrubs (BES) temperate'. SiB-table class 7 'groundcover' was used to parameterize the optical properties of grasses and crops ($\chi_L$ of -0.30). However, in later CLM versions such as in CLM5 (**Table 1.**), the optical properties of grass and crop PFTs were referenced to Asner et al. (1998), in which the estimates are presented only for spectral subsets following
different satellite sensor bandwidths (e.g. AVHRR bands 1 (VIS, 550-700 nm) and 2 (NIR, 725-1100 nm), and thus fail to represent full optical range. In addition, it is worth pointing out that the stem optical properties are defined based on dead leaves estimates reported by Dorman and Sellers (1989). Additional confusion may be caused by the fact that the SiB-table by Dorman and Sellers (1989) defines NIR region as 700-4000 nm, whereas in one of the SiB-table source datasets (in Sellers, 1985) the respective wavelength region is defined as 700-3000 nm. Noteworthy is that the current standard of measuring
spectral data extends only to 2500 nm. Although the spectral range used in this study does not cover the full theoretical range of total shortwave broadband albedo (300-4000 nm), the spectral range of 400-2400 nm contains ~96 % of the total solar irradiation (Thuillier et al., 2003, **Fig. 1.**) and thus suffices to approximate total VIS and NIR albedos. In addition, as the CLM-table contains a column for $\chi_L$, we assess their appropriateness based on measured and published data. In CLM5, the predefined angles ($\chi_L$, **Table 1.**) are 0.01 (~59.7°) for both NETs, 'NDT boreal', 'BDT tropical' and 'BES temperate', 0.10 (~56.6°) for
BETs, 0.25 (~51.3°) for BDT(/S) (refers to 'BDT+BDS') boreal and temperate, -0.3 (~69.5°) for grasses and C3 crops, and -0.5 (~75.5°) for other crops. As the focus of this paper is in optical properties, extensive review of leaf angle literature is not attempted.


**Table 1.** Collapsed version of the CLM5 optical properties table in CLM5 (2018), manual (Table 2.8). Note, in CLM the leaf angle value ($\chi_L$) is quantified based on divergence from spherical distribution: 1=planophile, -1=erectophile, and 0=spherical. Reflectance ($R$) and transmittance ($T$) in VIS (<= 700 nm) and in NIR (>=701 nm). 'BDT(/S)' contains both BDT and BDS PFTs for temperate and boreal PFTs.

| Plant Functional Type (PFT) | $\chi_L$ | $R$ (leaf, VIS) | $R$ (leaf, NIR) | $R$ (stem, VIS) | $R$ (stem, NIR) | $T$ (leaf, VIS) | T (leaf, NIR) | $T$ (stem, VIS) | $T$ (stem, NIR) |
|---|---|---|---|---|---|---|---|---|---|
| NET temperate; NET boreal; NDT boreal; BES temperate | 0.01 | 0.07 | 0.35 | 0.16 | 0.39 | 0.05 | 0.10 | 0.001 | 0.001 |
| BET tropical; BET temperate | 0.10 | 0.10 | 0.45 | 0.16 | 0.39 | 0.05 | 0.25 | 0.001 | 0.001 |
| BDT tropical | 0.01 | 0.10 | 0.45 | 0.16 | 0.39 | 0.05 | 0.25 | 0.001 | 0.001 |
| BDT(/S) temperate; BDT(/S) boreal | 0.25 | 0.10 | 0.45 | 0.16 | 0.39 | 0.05 | 0.25 | 0.001 | 0.001 |
| C3 arctic grass; C3 grass; C4 grass; C3 crop | -0.30 | 0.11 | 0.35 | 0.31 | 0.53 | 0.05 | 0.34 | 0.120 | 0.250 |
| Temp corn; Spring wheat; Temp soybean; Cotton; Rice; Sugarcane; Tropical corn; Tropical soybean | -0.50 | 0.11 | 0.35 | 0.31 | 0.53 | 0.05 | 0.34 | 0.120 | 0.250 |

## 2.2 Spectral databases

Spectral repositories used in this study are openly available online archives that were selected based on their reputation and methods used to collect the data (e.g. device, spectral range, metadata availability). To reduce differences in data resulting from different instrumentation, we only used leaf/needle-level data measured using integrating sphere to get both leaf $R_{(\lambda)}$ and

10    $T_{(\lambda)}$ information for forest and crop PFTs. For grasses 'canopy-level' $R_{(\lambda)}$ measurements were used (except for arctic grasses, for which the data were collected using leaf clip) (**Table 2.**).

The Lopex93 and Angers datasets belong to a group of 'Foundational datasets' defined as "Previously published spectroscopic data and associated metadata resources that represent exemplary, historically-notable, or transformational collections for the

15    environmental spectroscopy community" (EcoSIS, 2017). Lopex93 and Angers contains measurements of species with flat leaves, i.e., grasses, crops, broadleaved tree and shrub species. The Lopex93 campaign was organized by Joint Research Centre (JRC) in Italy during the summer of 1993 and contains leaf $R_{(\lambda)}$ and $T_{(\lambda)}$ data for leaves of 45 species. Angers is a dataset

collected by National Institute for Agricultural Research (INRA) in France on June 2003 containing $R_{(\lambda)}$ and $T_{(\lambda)}$ data for leaves from 39 species. For conifer needles few spectra are available due to obvious difficulties in measuring small needles. For boreal tree species two datasets are available by Lukeš et al. (2013) and Hovi et al. (2017) via SPECCHIO. For example, dataset by Hovi et al. (2017) contains $R_{(\lambda)}$ and $T_{(\lambda)}$ data for 25 Eurasian and North American boreal tree species measured

during peak-growing season. These two datasets contain measurements of both sun exposed and shaded leaves, by leaf sides (=adaxial and abaxial) from different canopy positions. For temperate conifers we used EcoSIS library from Serbin (2014) and data by Noda et al. (2014) stored in Japan Long-Term Ecological Research Network (JaLTER) archive. The dataset by Serbin (2014) was collected in the north-central and northeastern United States (US) as part of NASA Forest Functional Types Project (NNX08AN31G). The data from Noda et al. (2014) were measured in Japan with varying spectral ranges of 350-2500 nm and

350-2050 nm for foliage and bark (Note, spectra is available for leaves and shoots for different canopy positions; for foliage, the $R_{(\lambda)}$ and $T_{(\lambda)}$ are provided separately for abaxial and adaxial sides).

**Table 2. Spectral data used in this study. This table only lists data properties that were used in this study (i.e. the datasets may contain other data collected using different devices, methods or from different targets). Abbreviations: IS= integrating sphere, LC=**
**leaf clip, BF= bare optical fibre or optical head of spectrometer, $R_{(\lambda)}$= reflectance spectra, and $T_{(\lambda)}$= transmittance spectra. Target 'BDT(/S)' contains both BDT and BDS PFTs for temperate and boreal regions, and similarly 'BET(/S)' contains both BET and BES PFTs.**

| Reference | Name | Area | Target | Wavelength region | Geometry, and measurements |
|---|---|---|---|---|---|
| Jacquemound et al., (2003) | Angers 2003 | France | BDT(/S) temperate, BET(/S) temperate | 400-2450 nm | IS, $R_{(\lambda)}$ + $T_{(\lambda)}$ |
| Hoosgood et al., (1993) | Lopex93 | Italy | BDT(/S) temperate, BET(/S) temperate, Crops | 400-2500 nm | IS, $R_{(\lambda)}$ + $T_{(\lambda)}$ |
| Lukeš et al., (2013) | NA | Finland | NET boreal, BDT(/S) boreal | 400-2400 nm | IS, $R_{(\lambda)}$ + $T_{(\lambda)}$ |
| Hovi et al., (2017) | NA | Finland, Alaska | NET boreal, BDT(/S) boreal, NDT boreal | 350-2500 nm, 400-2300 nm | IS, $R_{(\lambda)}$ + $T_{(\lambda)}$ |
| Noda et al., (2014) | NA | Japan | NET temperate, NDT boreal; Bark | 350-2500 nm, 350-2050 nm | IS, $R_{(\lambda)}$ + $T_{(\lambda)}$, BF, $R_{(\lambda)}$ |
| Serbin, (2014) | NA | USA | NET temperate | 350-2500 nm | IS, $R_{(\lambda)}$ + $T_{(\lambda)}$ |
| Hall et al., (1996) | NA | USA | Bark | 350-2100 nm | BF, $R_{(\lambda)}$ |
| Lang et al., (2002) | NA | Estonia, Sweden | C3 grass, Bark | 400-2400 nm | BF, $R_{(\lambda)}$ |
| Toolik, (2017) | NA | Alaska | Arctic (C3) grass | 350-2500 nm | LC, $R_{(\lambda)}$ |
| Dennison and Gardner, (2018) | Hawaii 2000 | Hawaii | Tropical (C4) grass | 350-2500 nm | BF, $R_{(\lambda)}$ |

Bark $R_{(\lambda)}$ dataset was compiled using spectra from Noda et al. (2014), Hall et al., (1996), and Lang et al. (2002). In addition of containing bark $R_{(\lambda)}$ spectra, the Hall et al. (1996) dataset includes also measurements of branches, moss, and litter for boreal conditions (collected in Superior National Forest of Minnesota US). However, in this study we used dataset by Lang et al. (2002), to assess variation in $R_{(\lambda)}$ of different C3 grass compositions because the spectral range of data from Lang et al. (2002) was larger than that of Hall's data. For arctic (C3) grasses we used EcoSIS data measured in Toolik, arctic research field station in Alaska (Toolik, 2017). For tropical (C4) grasses we used EcoSIS data 'Hawaii 2000' dataset (Dennison and Gardner, 2018). In the absence of measured transmittance data for grasses, it was assumed equal (in **S1**) with that of crops (and grasses) contained by Lopex93 dataset.

## 2.3 Processing of the spectra

The spectra from different repositories were resampled to follow constant spectral range and interval (i.e. the spectral range and measurement interval of different devices varies and must therefore be unified). Spectra were resampled to have 1 nm interval within a spectral range of 400-2400 nm using R-package 'Prospectr' (Stevens and Ramirez-Lopez, 2015). The spectral regions with extreme noise were either removed or replaced with local means before smoothing. If 10% smoothing (span of 0.10) was enough to repair noisy regions in the spectra, no removals or replacements were done. Smoothing was done using loess regression (R default package) applying non-parametric least squares regression for localized subsets. Note, spectra >2400 nm was removed in effort to harmonize the spectral range of the different data sets (**Table 2.**). Normalized (i.e. to sum up to one) solar irradiance ($SI_{(\lambda)}$) spectra was used to weight both of $R_{(\lambda)}$ and $T_{(\lambda)}$ spectra before calculating the VIS (400-700 nm) and NIR (701-2400 nm) averages for $R$ and $T$ (i.e. all band averages of $R_{(\lambda)}$ and $T_{(\lambda)}$ are given in after weighting with $SI_{(\lambda)}$). We used the white-sky $SI_{(\lambda)}$ measured at sea-level to account for atmospheric scattering and absorption effects (**Fig. 1.**). The $SI_{(\lambda)}$ was normalized (i.e. to sum up to 1), separately for VIS and NIR wavelength bands (i.e. the relative shape of the $SI_{(\lambda)}$ within the VIS and NIR subset was preserved). Foliage $SSA_{(\lambda)}$ was obtained as a sum of $R_{(\lambda)}$ and $T_{(\lambda)}$ (separately for VIS and NIR) and multiplied with the respectively (i.e. VIS or NIR) normalized $SI_{(\lambda)}$. The leaf or needle $SSA$ was obtained as a sum over the resulting VIS and NIR bands. The $SI_{(\lambda)}$ normalization was adapted to shorter NIR spectral ranges of Hall et al. (1996), Hovi et al. (2017) and Noda et al. (2014) data (in **Table 2.**). All VIS and NIR -averaged $R$, $T$ and $SSA$ values represented in this paper have been weighted with the $SI_{(\lambda)}$ to show results consistently.

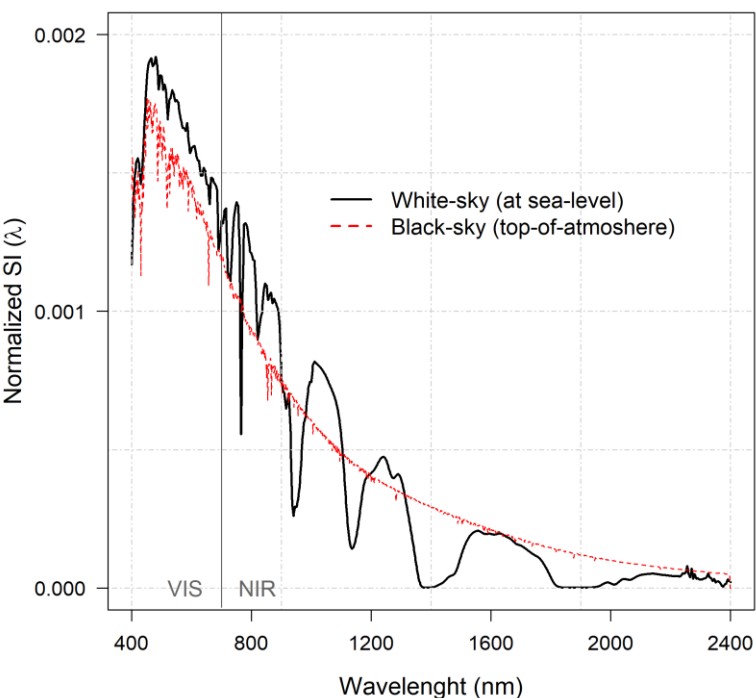

Figure 1. The normalized (i.e. sums to unity between 400-2400 nm) white-sky solar irradiance spectra ($SI_{(\lambda)}$) measured at sea-level, and for a reference, the normalized black-sky top-of-the-atmosphere spectra by Thuillier et al. (2003). Figure is shown to illustrate the effect of atmosphere on the shape of $SI_{(\lambda)}$ (Note, in our calculation the white-sky spectra was renormalized within VIS (400-700 nm) and NIR (701-2400 nm) subsets.

## 2.4 Upscaling spectra from needle to shoot

Clustering of needles into shoots causes the $R$ and $T$ of shoots to be systematically smaller than that of needles, due to within-shoot multiple scattering (Stenberg, 1996). By replacing the VIS and NIR $SSA_{needle}$ with $SSA_{shoot}$, the systematic bias caused by shoot-level clumping can be accounted for in RT modeling. The spectra of needles can be upscaled to shoot-level using spherically averaged Silhouette to Total needle Area Ratio ($STAR$, e.g. Oker-Blom and Smolander, 1988; Stenberg, 1996). For a shoot without within-shoot shadowing, the $STAR$ would be 0.25 because the spherically averaged projection area of a convex needle is one fourth of its total area (Lang, 1991). The $STAR$ is known to vary between species and canopy positions (and may vary e.g. from 0.12 to 0.28), and in the absence of adequate data the $STAR$ can be approximated using a value of 0.16 for a range of shoot structures (Thérézien et al., 2007). In this study a constant $STAR$ of 0.16 was used for all conifer species for demonstration. At shoot-level the $p$ is linearly related with $STAR$ (i.e. $p = 1 - 4 \times STAR$ under diffuse radiation conditions), which allows upscaling the $SSA(\lambda)$ (i.e. $SSA_{needle(\lambda)} = R_{(\lambda)} + T_{(\lambda)}$) to $SSA_{shoot(\lambda)}$ (Smolander and Stenberg, 2003; Rautiainen et al., 2012) as following Eq. (1):

$$SSA_{shoot(\lambda)} = SSA_{needle(\lambda)}\left(\frac{1-p}{1-p\,SSA_{needle(\lambda)}}\right) \tag{1}$$

The $SSA_{shoot(\lambda)}$ were multiplied with normalized $SI_{(\lambda)}$ for VIS and NIR wavelength regions as explained in **section 2.3**, and
$SSA_{shoot}$ in VIS and NIR were obtained by taking the sum over the spectra. Note, when *STAR* is greater than 0.25, then $SSA_{shoot}$ > $SSA_{needle}$, which may happen if shoot structure is abnormal (e.g. shoot has very short needles which only cover the upper side of the twig (Thérézien et al., 2007).

## 2.5 Leaf angle specification

Leaf angle distribution (LAD) of foliage determines radiation transmission though plant canopies and is also included in the CLM5-table (in a form of $\chi_L$). The assumption on random foliage distribution remains valid for many conifer species (e.g. Barclay, 2001), and thus we focus on providing some example data for other PFTs. The leaf angle properties of PFTs will be defined based on data presented in Wang et al. (2007), Chianucci et al., (2018), Pisek et al., (2011), Gratani and Bombelli, (2000), Zou et al., (2014), and Ross (1981).

Wang et al. (2007) reported measured *LIA*s for three grass species (i.e. *Andropogon gerardii*, *Panicum vigratum*, and *Sorghastrum nutans*) measured in Konza Prairie in North America (data from Li, 1994), and for leaves of 38 species including flowering plants, shrubs and trees measured in Ku-ring-gai Chase National Park in Australia (data from Falster and Westoby, 2003). R-library 'RLeafAngle' (Wang et al. (2007), contains dataset called 'Falster' for 38 species published by Falster and
Westoby (2003). The Falster data were used to obtain *LIA* estimates for 'BDT(/S) tropical'. For 'BDT(/S) temperate' and 'BDT(/S) boreal' *LIA* estimates were obtained from data by Chianucci et al. (2018), which contains *LIA* measurements for 55 tree and shrub species collected by Pisek et al. (2013) and Raabe et al. (2015) at different sites in Sweden, Estonia, and the USA, and for 83 species at various sites in central Italy. The mean *LIA* of 'BET(/S) temperate' and 'BET(/S) tropical' was approximated using two species from Hawai'i (i.e. *Metrosideros polymorpha*, and *Schizostachyum glaucifolium* (=bamboo)
in Pisek et al. (2011) and two from the Mediterranian region (i.e., *Phillyrea latifolia*, and *Quercus ilex*, in Gratani and Bombelli (2000)).

The variety of mean $\chi_L$ or *LIA* estimates of different grass and crop species was demonstrated using data compiled by Ross (1981) from various sources (Note, **S4** contains a .pdf copy of the leaf angle tables presented by Ross (1981) and references to
original works), Li (1994), and Zou et al. (2014). As it is not easy to classify different plant species into either crop or grass, we chose to present these data by dataset (see **S3** for details). The pure 'Grasses' (Li, 1994) and cool-temperate 'Crops' (Zou

et al., 2014) data contained measured *LIA* estimates, but 'Crops+Grass' data reported values using $\chi_L$ which were converted to *LIA* in degrees (°). The conversion between mean *LIA* ($\theta_{mean}$) and $\chi_L$ was approximated following CLM5 (2018) manual as:

$$\cos\theta_{(mean)} = \frac{1+\chi_L}{2} \qquad\qquad (2)$$

For each PFT the mean *LIA* estimates in degrees and as dispersion from spherical ($\chi_L$) were obtained as an average across species means (species level data listed in **S3.**). The species mean *LIA* estimates were assigned to classic LAD types (de Wit, 1965) using RLeafAngle -package function 'selectClassic()' and thus may differ from that presented in the original works.

## 3 Results

### 3.1 Forest PFTs

#### 3.1.1 Optical properties of forest PFTs

The reflectance ($R$) and transmittance ($T$) of conifer needles were similar between 'NET temperate' and 'NET boreal' in both VIS and NIR wavelengths (**Fig. 2, Table 3.**). For example, for 'NET temperate' mean $R_{VIS}$ was 0.08, and mean $T_{VIS}$ was 0.04, and for 'NET boreal' the respective values were 0.09 and 0.05. Similarly, for 'NET temperate' ('NET boreal') the mean $R_{NIR}$ was 0.41 (0.41) and mean $T_{NIR}$ was 0.31 (0.33). Thus, the CLM default $R_{VIS}$ and $T_{VIS}$ (0.07, 0.05) for NET appear appropriate (**Table 1., Fig.2a**). However, the CLM default $R$ and $T$ in NIR are not at the correct level: the CLM defaults for $R_{NIR}$ and $T_{NIR}$ are 0.35 and 0.10 - but based on our data the values should be ~0.41 and ~0.32, respectively (**Fig. 2a, Fig. 3a**).

The mean $R$ and $T$ were similar also for temperate and boreal BDTs (**Table 3.**). For 'BDT temperate', the mean $R_{VIS}$ was 0.08 and mean $T_{VIS}$ was 0.06, and for 'BDT boreal', the respective values were 0.09 and 0.05. Similarly, for 'BDT temperate' the mean $R_{NIR}$ was 0. 42 and mean $T_{NIR}$ was 0.43, while for 'BDT boreal' the respective values were 0.40 and 0.42 (**Fig. 3b**). Thus, we can conclude that the CLM for BDT $R_{VIS}$ and $T_{VIS}$ are appropriate ($R_{VIS}$ = 0.10 and $T_{VIS}$ = 0.05) (**Fig. 2b**). However, the CLM default value for 'BDT temperate and boreal' $T_{NIR}$ of 0.25 requires an update: based on our data the respective $T_{NIR}$ should be and ~0.43. For $R_{NIR}$ the CLM default is 0.45 and the mean measured value was ~0.41.

For BET temperate, the averages of $R_{VIS}$ and $T_{VIS}$ were 0.11 and 0.06, respectively (**Fig. 3c**, **Table 3.**). These values corresponded well with the CLM default values (i.e. $R_{VIS}$= 0.10 and $T_{VIS}$= 0.05). However, the CLM default $T_{NIR}$ of 0.25 was slightly smaller than the mean measured $T_{NIR}$ of 0.33) **(Fig. 2b)**. The CLM default BET $R_{NIR}$ of 0.45 correspond well with the measured mean $R_{NIR}$ of 0.46.

Results showed that the CLM optical properties for 'NDT boreal' are fine in VIS (**Fig. 2a**). However, CLM defaults for $R_{NIR}$ and $T_{NIR}$ of 0.35 and 0.10, were found too low – The $R_{NIR}$ and $T_{NIR}$ should be ~0.39 and ~0.42 based on measured data (**Fig. 3d**). Noteworthy is that, the 'NDT boreal' optical properties are more similar in NIR with BDT than with NET, which is the CLM default grouping (**Fig. 2ab.**).

Based on our calculation, the $SSA_{shoot}$ are on average 32% smaller in VIS and 10% smaller in NIR than the $SSA_{needle}$ (**Table 3.**, **S2**). The largest differences between the two albedo proxies resulted in a 36% difference in VIS and 15% difference in NIR, with the smallest differences being 29% in VIS and 7% in NIR.

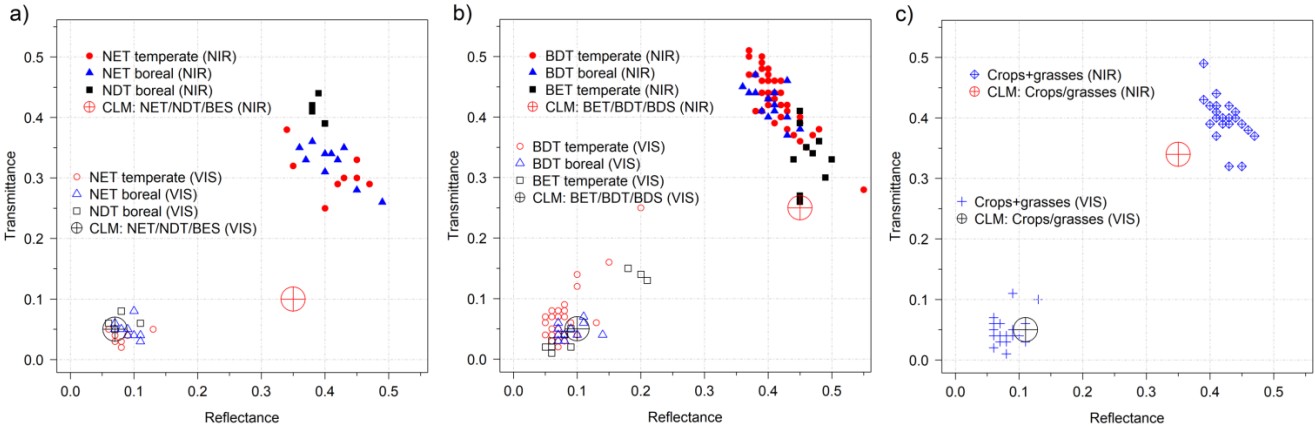

**Figure 2. Leaf reflectance and transmittance in VIS (400-700 nm) and NIR (701-2400 nm) for different PFTs. Single species values are plotted to demonstrate the within- and between- PFT variation for a) 'NET temperate' and 'NET boreal', 'NDT boreal', b) 'BDT temperate' and 'BDT boreal', and 'BET temperate', and c) 'Crops+grasses'. The CLM default optical properties are plotted using large symbols. For colors, please see the online version of the article.**

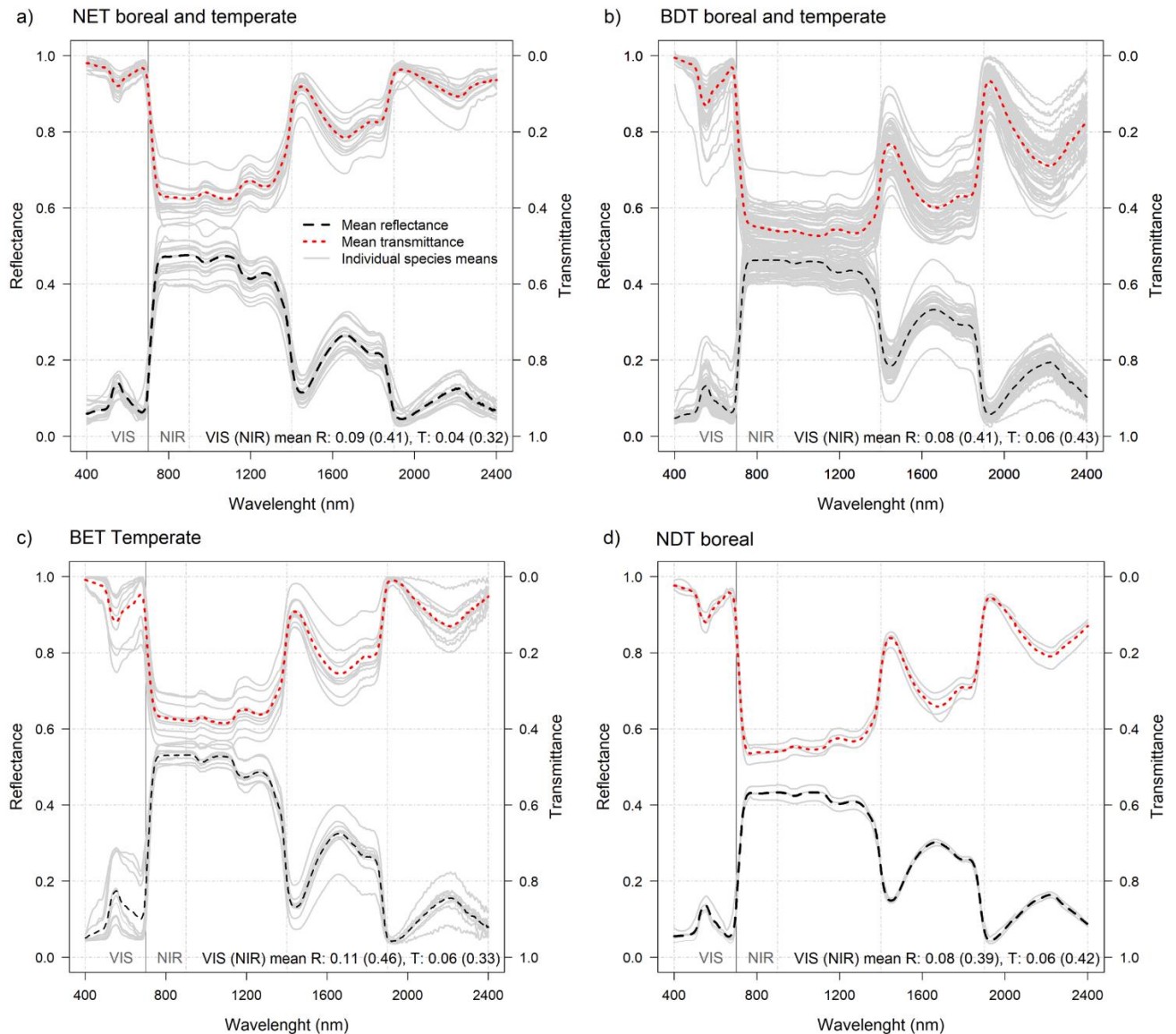

**Figure 3. Average reflectance ($R_{(\lambda)}$) and transmittance ($T_{(\lambda)}$) spectra of foliage for forest Plant Functional Types (PFTs) for *n* species samples: a) NET boreal and temperate (n=18), b) BDT boreal and temperate (n=54), c) BET temperate (n=10), and d) NDT boreal (n=4). Individual species mean $R_{(\lambda)}$ and $T_{(\lambda)}$ spectra are shown using light grey lines and are used to represent the within PFT deviation in mean spectra (see individual species estimates in S2). The mean $R_{(\lambda)}$ and $T_{(\lambda)}$ for VIS (400-700 nm) and NIR (701-2400 nm) wavelengths are provided at the bottom of the images and are calculated as an average of the species means. Note, all VIS and NIR averages of $R_{(\lambda)}$ and $T_{(\lambda)}$ were weighted with solar irradiance spectra ($SI_{(\lambda)}$).**

**Table 3. Average leaf or needle reflectance (*R*), transmittance (*T*), single scattering albedo (*SSA*) and single scattering albedo corrected for shoot-level clumping (*SSA*$_{shoot}$) over VIS (400-700 nm) and NIR (701-2400 nm) wavelength regions by Plant Functional Type (PFT). Note, all VIS and NIR averages of *R*$_{(\lambda)}$ and *T*$_{(\lambda)}$ were weighted with solar irradiance spectra (SI$_{(\lambda)}$). Standard error inside parenthesis.**

| PFT | $R_{VIS}$ | $T_{VIS}$ | $SSA_{VIS}$ | $SSA_{shootVIS}$ | $R_{NIR}$ | $T_{NIR}$ | $SSA_{NIR}$ | $SSA_{shootNIR}$ |
|---|---|---|---|---|---|---|---|---|
| NET temperate | 0.08 (0.02) | 0.04 (0.01) | 0.12 (0.02) | 0.08 (0.02) | 0.41 (0.05) | 0.31 (0.04) | 0.72 (0.04) | 0.64 (0.05) |
| NET boreal | 0.09 (0.02) | 0.05 (0.01) | 0.14 (0.01) | 0.1 (0.01) | 0.41 (0.04) | 0.33 (0.03) | 0.74 (0.03) | 0.66 (0.03) |
| NDT boreal | 0.08 (0.02) | 0.06 (0.01) | 0.15 (0.02) | 0.1 (0.02) | 0.39 (0.01) | 0.42 (0.02) | 0.81 (0.02) | 0.74 (0.02) |
| BET temperate | 0.11 (0.06) | 0.06 (0.06) | 0.17 (0.12) | | 0.46 (0.02) | 0.33 (0.05) | 0.8 (0.05) | |
| BDT temperate | 0.08 (0.03) | 0.06 (0.04) | 0.14 (0.07) | | 0.42 (0.03) | 0.43 (0.05) | 0.84 (0.03) | |
| BDT boreal | 0.09 (0.02) | 0.05 (0.01) | 0.14 (0.02) | | 0.4 (0.03) | 0.42 (0.03) | 0.83 (0.02) | |
| Crop | 0.08 (0.02) | 0.05 (0.02) | 0.13 (0.03) | | 0.42 (0.02) | 0.4 (0.04) | 0.82 (0.03) | |

### 3.1.2 Optical properties of tree bark

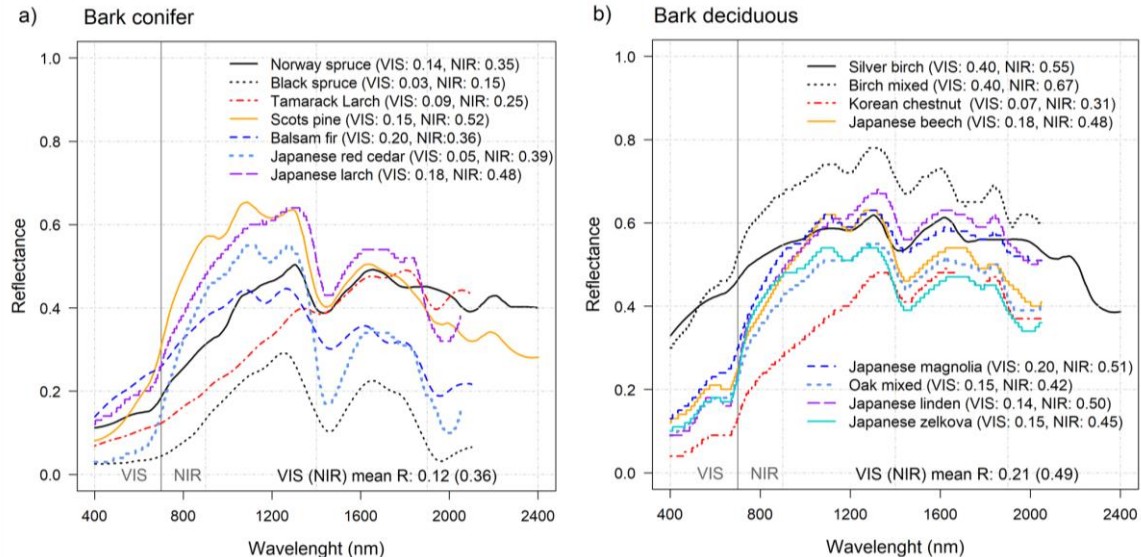

**Figure 4. Bark reflectance (*R*$_{(\lambda)}$) of a) coniferous, and b) deciduous species in VIS (400-700 nm) and NIR (701-2400 nm) wavelength regions. The mean bark reflectance values in VIS and NIR are provided at the bottom of the images and are calculated as an average**

10 **across species means (the mean across all bark values *R*$_{VIS}$ =0.17 and *R*$_{NIR}$ = 0.43). Note, all VIS and NIR averages of *R*$_{(\lambda)}$ and *T*$_{(\lambda)}$ were weighted with solar irradiance spectra (SI$_{(\lambda)}$). For colors, please see the online version of the article.**

Based on our data sample, coniferous bark $R_{VIS}$ varied between 0.03 and 0.20 and in $R_{NIR}$ between 0.15 and 0.52 (**Fig. 4a, S2**). The average coniferous bark $R_{VIS}$ was 0.12 and for $R_{NIR}$ 0.36. For deciduous species, the bark $R_{VIS}$ varied between 0.07 and

15 0.40, and in $R_{NIR}$, between 0.31 and 0.67 (**Fig. 4b**). The average deciduous species bark $R_{VIS}$ and $R_{NIR}$ were 0.21 and 0.49, respectively. In CLM, the same constant stem reflectance is used for all forested PFTs ($R_{VIS}$ of 0.16 and $R_{NIR}$ of 0.39). Thus,

the CLM default bark reflectance in VIS and NIR fall within the range of measured values (average over all species $R_{VIS} = 0.17$ and $R_{NIR} = 0.43$). However, alternatively bark reflectance could be defined separately for coniferous and deciduous PFTs.

## 3.2 Optical properties of grass and crop PFTs

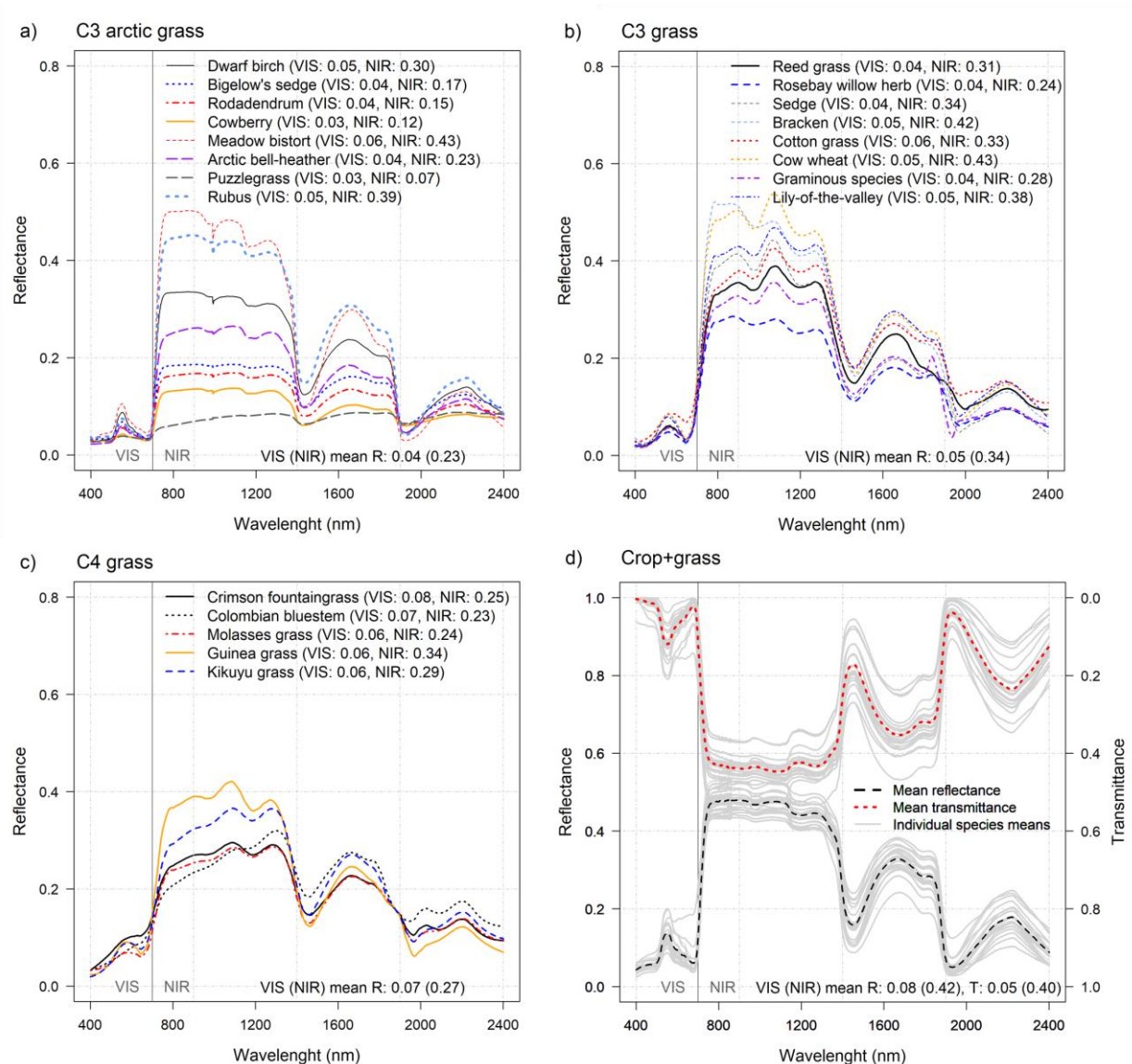

5 **Figure 5. Optical properties of grasses and crops.** Reflectance ($R_{(\lambda)}$) of: a) C3 arctic grasses, b) C3 grass canopies, c) C4 grass canopies, and d) and transmittance ($T_{(\lambda)}$) of leaves of different crops (n=21, contains also some grass species): Individual species mean $R_{(\lambda)}$ and $T_{(\lambda)}$ are shown using light grey lines and are used to represent within PFT deviation in mean spectra (see individual species values in S2). The mean $R$ and $T$ for VIS (400-700 nm) and NIR (701-2400 nm) wavelengths are provided at the bottom of the images and are calculated as an average across species means. Note, all VIS and NIR averages of $R_{(\lambda)}$ and $T_{(\lambda)}$ were weighted with solar irradiance

10 spectra ($SI_{(\lambda)}$). For colors, please see the online version of the article.

Leaf reflectance spectra for different grass species demonstrates large within-PFT variation, which exceeds the differences between various grass PFTs (i.e. C4, C3 arctic, and C3 grasses). The leaf mean $R_{VIS}$ ($T_{VIS}$) of different grass and crop types (i.e. C3 arctic grass, C3 grass, C4 grass, and crops were: 0.04 (0.23), 0.05 (0.34), 0.07 (0.27), and 0.08 (0.42), respectively (**Fig. 5.**, **S2**). In the CLM table, the leaf default $R_{VIS}$ and $R_{NIR}$ are 0.11 and 0.35 for all grass and crop PFTs. The CLM default leaf $R_{VIS}$ seems a little high as only 3/42 grass or crop species (i.e. garden lettuce, corn and soybean) reach the $R_{VIS}$ of 0.11. The $R_{NIR}$ of 0.35 on the other hand stands out like an outlier (**Fig. 2c.**, **S2**), and thus slightly higher value could be used. For crops, the measured mean leaf $T_{VIS}$ and $T_{NIR}$ was 0.05 and 0.40, respectively. Thus, although the measured leaf $T_{VIS}$ values aligned perfectly with the CLM default value (of 0.05), the CLM default leaf $T_{NIR}$ value of 0.34 needs an update. For grasses the updated leaf $R_{VIS}$ and $R_{NIR}$ could be ~0.05 and ~0.28, and for crops ~0.08 and ~0.42 (**S2**). In the absence of measured transmittance data for grasses, the $T_{VIS}$ and $T_{NIR}$ of grasses could be defined based on respective crop values (i.e. 0.05 and 0.40).

### 3.3 Leaf angle specification

Based on measured data, the mean *LIA* of 'BDT tropical' ($\chi_L$ of 0.20 i.e. ~52.1°) was found more planophile than what is the CLM default value of 0.01 ($\chi_L$, i.e ~60°) (**Table 4, S3**). However, as there is a lot of variation among *LIA* estimates between species (i.e. $\chi_L$ ranges from -0.42 to 0.84), the assumption on spherical foliage orientation seems fine for 'BDT(/S) tropical'. For 'BDT(/S) temperate/boreal' the mean *LIA* across species means was 36.0° (i.e. $\chi_L$ of ~0.59) and, thus was also found more planophile than the CLM5 default of ~51.3° (i.e. $\chi_L$ of 0.25). Consequently, the $\chi_L$ value of 'BDT(/S) temperate/boreal' could be adjusted to correspond better with observed variation in the data. For 'BET(/S) temperate/tropical' the mean *LIA* was 48.5° (i.e. $\chi_L$ of 0.32) and thus somewhat agreeing with the CLM5 default of ~56.6° ($\chi_L$ of 0.10).

**Table 4. Mean leaf inclination angles (*LIAs*) of different flat-leaved Plant Functional Types (PFTs). The angles are provided both in degrees (°) and as departures from spherical ($\chi_L$). Number of observations is shown in column '*n*'. Individual species estimates are presented in supplement S3.**

| PFT | Mean(°) | Sd(°) | Min(°) | Max(°) | Mean($\chi_L$) | Sd($\chi_L$) | Min($\chi_L$) | Max($\chi_L$) | n |
|---|---|---|---|---|---|---|---|---|---|
| BDT(/S) tropical | 52.1 | 12.4 | 23.2 | 73.0 | 0.20 | 0.33 | -0.42 | 0.84 | 38 |
| BDT(/S) temperate/boreal | 36.0 | 10.9 | 12.9 | 69.4 | 0.59 | 0.24 | -0.30 | 0.95 | 138 |
| BET(/S) temperate/tropical | 48.5 | 6.2 | 43.5 | 57.1 | 0.32 | 0.17 | 0.09 | 0.45 | 4 |
| Grass | 67.4 | 5.5 | 61.3 | 72.7 | -0.23 | 0.18 | -0.41 | -0.04 | 5 |
| Crops (cool-temperate) | 41.2 | 18.3 | 17.6 | 63.2 | 0.44 | 0.41 | -0.10 | 0.91 | 6 |
| Crops+Grass | 51.0 | 6.0 | 39.7 | 60.7 | 0.25 | 0.16 | -0.02 | 0.54 | 18 |

For the non-forest PFTs (i.e. grasses and crops), the CLM5 default parameterization of $\chi_L$ was either -0.30 (~69.5°) or -0.50 (~75.5°) depending on vegetation type. Based on measured data the mean $\chi_L$ of grasses (of -0.23, ~67.4°) was found to

corresponding well with the CLM5 default value. However, for crops the observed $\chi_L$ values were clearly leaning towards more planophile (e.g. 41.2° and 51.0°, $\chi_L$ of 0.44 and 0.25) than erectophile foliage orientation (i.e. ~75.5°, $\chi_L$ of -0.5). Cool-temperate crops demonstrated the largest variation in *LIA*s (ranged from 17.6º to 63.2º, i.e. $\chi_L$ from of -0.10 to 0.91). Noteworthy is that from among 29 grass and crops species, none reached $\chi_L$ of -0.50; however, two grasses had $\chi_L$ of -0.40

(**Table 4**, **S3**). Thus, based on the data shown in this study, the CLM5 default $\chi_L$ of crops should be updated. The mean $\chi_L$ of the crop species presented here was 0.30 (~48.5º) (**S3**).

## 4. Discussion

Based on a dataset compiled following a synthesis and harmonization of spectral data found in a variety of data repositories,
we showed that many optical properties based on the 'SiB-table' (currently used by e.g. CLM) are in need of an update. While the optical properties were by default at the correct level in VIS wavelength region (determines vegetation productivity *via* photosynthesis), the changes in optical properties in the NIR wavelength region may be expected to have an impact on predicted surface albedo. To our knowledge, only Göttlicher et al. (2011) have made an attempt to verify the CLM optical parameters of 'BET tropical' (PFT) using measured spectral data. However, as their NIR data covered only a part of the
spectrum (from 701 -1300 nm), only VIS verification was obtained. We cannot argue that the values presented in this paper are the 'truth' *per se*, nor that researchers should use the values presented in this paper. However, we can state that there are systematic biases in the optical property values in the NIR wavelength region, across all PFTs. For example, for NET and NDT, the empirically based $SSA_{needle}$ values exceeded the CLM default parameters by 62% and 78%, respectively - even after accounting for shoot-level clumping, the $SSA_{shootNIR}$ were still 44.4% (NET) and 64.4% (NDT) larger than the CLM defaults.
Similarly, for the BDT, BET and crop PFTs, the measured leaf $SSA_{NIR}$ values were 20.0%, 14.3% and 18.8% larger than the CLM default estimates, respectively (numbers calculated based on **S2**). Noteworthy is that, as LSMs are often run using PFT distributions obtained from remotely sensed landcover products, and as there are no possibilities for within-PFT species differentiation, the use of constant shoot-structural factor to upscale $SSA_{needle}$ to $SSA_{shoot}$ may be justified. However, for other applications having species information readily available, the species-specific shoot structural factors should be used.
According to Rautiainen et al. (2012), $SSA_{shoot}$ are considerably smaller than $SSA_{needle}$, and there is more variation in shoot spectra (coefficient of variation, CV, 8-21%) than in the needle spectra (CV 2-13%) due to the geometry of the shoot. In this study, the $SSA_{shoot}$ in VIS and NIR were ~30% and ~10% smaller, respectively, than the $SSA_{needle}$ (note that a constant factor was used).

As optical properties represent the effective surface variables, we can argue that there is a need to update the parameters, as changes in initial parameterization may be expected to result in changes in predicted surface albedo. However, whether or not (and if yes, then to what extent) changes in optical properties result in changes to predicted surface albedo requires LSM simulations since LSMs have been tuned to reduce influences of identified biases and possible compensating errors. For

example, in the case of CLM, no global soil reflectance dataset was available during model development and soil reflectance data is now based on the tuning of the CLM simulated surface albedo to match MODIS observations (Lawrence and Chase, 2007). While this may not be too important in dense canopies with high LAI, in sparse canopies (LAI < 2) soil reflectance becomes more important. In addition, we must consider that the major assumptions of 1D RT models themselves are likely to

source some error: CLM employs a simplified plane-parallel, two-stream model based on the homogenous turbid medium assumption with isotropic scattering properties. 1D models commonly ignore stems and branches, but the stems are accounted by the CLM RT model: Optical parameters are calculated as a weighted average of leaf and stem areas (i.e. LAI and SAI). This may introduce possible errors (or biases), because **i)** empirical data and theoretical basis for more accurate definition of SAI is currently lacking (e.g. CLM5 manual, section 29.5.2 (CLM5):" The existing CLM(CN) algorithm sets the minimum

SAI at 0.25 to match MODIS observations, but then allows SAI to rise as a function of the LAI lost, meaning than in some places, predicted SAI can reach value of 8 or more. Clearly, greater scientific input on this quantity is badly needed."); and **ii)** incompatibilities in vegetation structural descriptions in employed RT models (i.e. MODIS LAI is based on 3D RT model whereas CLM employs 1D RT model), which may lead to erroneous assessments of the absorbed, transmitted and reflected fluxes (Pinty et al., 2004; 2006)). Noteworthy is that we did not provide updated optical property values for the stems of grasses

and crops due to the scarcity of measured spectral data for these plant components. However, considering that information on SAI is currently lacking, and that grass and crop stems are ignored by today's RT models employed in vegetation remote sensing applications (e.g. MODIS LAI algorithm (Knyazikhin et al., 1999) and PROSAIL (Jacquemoud et al., 2000)), optical properties of grass and crops stems could also be ignored in CLM RT simulations to correspond better with MODIS LAI.

Generally speaking, the need for improving the RT models employed in LSMs has been acknowledged, and progress has already been made (Yuan et al., 2017; McGrath et al., 2016). For example, a 'domain-averaged structural factor' (i.e. effective LAI accounting for inhomogeneous horizontal distribution such as tree clumping and canopy gaps) and multilayer canopy vertical albedo profile were recently added by McGrath et al., (2016) for ORganising Carbon and Hydrology In Dynamic EcosystEms (ORCHIDEE, SVN r2566) model. In their approach, tree crowns were treated as spheroids filled with turbid

medium with infinitely small scatterers, and tree trunks were ignored as spectral parameters are extracted from remote sensing data without differentiation between leafy and woody areas. However, as pgap model (Haverd et al., 2012) accounts tree trunks in canopy gap parameterization, the trunks should ideally also be accounted as canopy spectral parameters are determined (Naudts et al., 2015; McGrath et al., 2016). They modeled grasses and crops as homogenous blocks, without internal structure, and defined tunable 'correction factor' to account clumping effects. For CLM, recent advancements were done by Yuan et al.,

(2017) who compared four representative 1D RT models under the same framework and implemented the appropriate modifications for the CLM4.5 (Oleson et al., 2013). They proposed changes for the employed LAD equation, and two modifications following paper by Pinty et al., (2006) regarding the treatment of incident diffuse radiation and backward scattering coefficient for indecent direct radiation.

As an alternative for empirically based correction factors, which may potentially violate the law of energy conservation, new perspectives for the old challenge are offered by theory of spectral invariants: Based on spectral invariants theory, $SSA(\lambda)$ is the only parameter that depends on wavelength, while all other parameters are determined by canopy structural factors (Wang et al., 2018). In this paper, we demonstrated how information on shoot-geometry (i.e. $p$) can be used to upscale $SSA_{needle(\lambda)}$ into

effective $SSA_{shoot(\lambda)}$ to account for within-shoot scattering, which violates the basic assumptions behind the RT calculation (i.e. non-random ordering of finite-sized needles). The proposed correction, is not currently accounted for in LSMs, and **i)** can be incorporated by simply replacing $SSA_{needle}$ with effective $SSA_{shoot}$ in the RT calculation, **ii)** is applicable to RT models employing turbid media assumption and Beer's law, and **iii)** provides simplicity required by LSMs. In addition of spectral invariants theory being already incorporated into MODIS LAI algorithm (Knyazikhin et. al., 1999), other desirable features

from the point of LSM are that $p$: **i)** allows generation of consistent products from satellite sensors operating at different spatial resolutions (Ganguly et al., 2008a), and **ii)** permits compressing 3D information into 1D form across various spatial domains (Ganguly et al., 2008b), and **iii)** allows measuring, scaling and validation (Stenberg et al., 2016). As remotely sensed products are used as an input in LSMs, advances in RT modeling employed in remote sensing should ideally be reflected by LSM RT parameterizations. In addition, more effort in LSM RT modeling is needed for developing scaling routines to account for

seasonal changes of optical properties (and SAI), and for improving parameterizations for snow and ice (Yuan et al., 2017).

PFT definitions are needed by LSMs to classify species into groups of similar structural and functional characteristics. While that appears a relatively simple task, this is not always the case. For example, while the difference between tree and a shrub might seem easy to define, in practice defining these two is complicated by overlapping definitions. While both trees and

shrubs are perennial woody plants, a shrub is considered shorter in stature than a tree and typically has more stems. However, a shrub may have as few as one stem and be tall in stature (up to 3 or 4 meters in height) analogous to a small tree. Thus, the optical properties of shrub PFTs could be defined based on respective forest PFTs optical properties. In practice, we suggest that optical properties of e.g. 'BES temperate' be based on 'BET tropical' and 'BET temperate' instead of on 'NET temperate/boreal' and 'NDT boreal' optical properties, which is the default CLM grouping. In addition, as the optical properties

of the 'NDT boreal' are more like those of BDT (especially in NIR) than to NET which is the current CLM default grouping - the CLM could classify NDT into BDT group rather than NET. Further, pending the PFT-boundaries: Here we classified English ivy as belonging to 'BET(/S) temperate/boreal', despite it being an evergreen vine, and bamboo as 'BET(/S) temperate/tropical' (as it can reach up to 15 meters height and has flat-leaf structures), although it is flowering plant rather than a tree or shrub. In addition, dataset by Chianucci et al. (2018) contained two plants belonging to the grass family and

one fern species, but as their growth form was recorded as tree or shrub, we decided to keep them in data.

Many of today's land surface models such as JSBACH (JSBACH, 2019), JULES, and ORCHIDEE assume LAD to be spherical. However, the assumption of spherical LAD has been found to cause significant underestimation of light transmission (Stadt and Lieffers, 2000), and has been found to be invalid for most temperate and boreal deciduous tree species based on an

extensive dataset of measured LADs (Pisek et al., 2013; Chianucci et al., 2018) (e.g. only 14 of 138 species the LAD was spherical). Another study with Australian species showed that only 3/12 types of herbaceous plant canopies and 8/38 plant species (e.g. trees, woody shrubs, climbers, ferns and cycads) had spherical LAD (Wang et al., 2007). Note that these two datasets were used also in this study. In CLM the LAD definition denotes the departure from spherical: Based on results the

CLM default LAD definition of forested PFTs could be slightly more planophile. For LSMs which can implement non-spherical LAD definitions, LAD parameters for a range of species are readily available from Chianucci et al. (2018) and Wang et al. (2007). However, the finding that CLM5 default LAD for crops is notably too vertical (i.e. CLM5 default crop $\chi_L$ stands out as an outlier from among the empirical observations) requires attention from modelers. We acknowledge, that while LAD may be assumed a species-specific parameter, it may be hard to estimate correctly as it changes based on plant development

stage (e.g. crops such as Maize (see S4)) and as a response to solar illumination conditions (i.e. dual role of being exposed for solar radiation to enable photosynthesis, but to avoid overexposure which would cause heat stress). Thus, future studies are needed to address the issue of the PFT LAD definition, especially in the case of grasses and crops that are more exposed to solar radiation than trees. As an alternative to field measurements, LAD may be also inverted based on remotely sensed data (Huang et al., 2006).

In the future, large databases which systematically collect chemical and spectral data at different scales (i.e. from leaf to canopy-level) and standardized protocols for field and lab work may be expected to become more common (e.g. Asner and Martin, 2016). While the motivation of remote sensing scientists is to build these databases to foster scientific discoveries, the same databases could also be used to provide inputs for different LSMs (especially those employing plant traits). The build-

up of larger databases would solve most of present-day problems in terms of data usability by providing standardized data access policies, data formats, preprocessing, and metadata. We should aim for truthful description of vegetation properties in different LSMs, as that is prerequisite for increasing the accuracy of the predictions.

**Conclusions**

Using the CLM PFT grouping as an example, we found that the default PFT optical parameters fell within the range of measured values in VIS band, but in the NIR band updates are needed. Such updates may be expected to have direct impact in the modeling of surface albedo and the shortwave radiation balance, and in turn, fluxes of $CO_2$, moisture, and energy at the surface. Thus, we encourage modelers employing two-stream RT approximations based on leaf-level optical properties to check their models' default optical property parameters and consider using shoot-level clumping corrected values for NET and

NDT.

**Code availability**

No code available for this manuscript.

**Data availability**

Leaf angle dataset by Falster and Westoby, (2003) is available via RLeafAngle R-package (dataset name "Falster"), and data from Chianucci et al., (2018) from https://data.mendeley.com/datasets/4rmc7r8zvy/2. Optical property estimates calculated from the raw data are included in SI. Raw spectral data is stored into openly available data repositories (listed in **Table 2**):

Jacquemound et al., (2003):

https://ecosis.org/#result/2231d4f6-981e-4408-bf23-1b2b303f475e

Hoosgood et al., (1993):

https://ecosis.org/#result/13aef0ce-dd6f-4b35-91d9-28932e506c41

Lukeš et al., (2013):

Dataset name "OP_measurements" available via https://specchio.ch/

Hovi et al., (2017):

Dataset name "Hovi_et_al_2017_Silva_Fennica(Version1.0)" available via https://specchio.ch/

Noda et al., (2014):

http://db.cger.nies.go.jp/JaLTER/metacat/metacat?action=read&qformat=default&sessionid=&docid=ERDP-2013-02.1

Serbin, (2014):

https://ecosis.org/#result/4a63d7ed-4c1e-40a7-8c88-ea0deea10072

Hall et al., (1996):

https://daac.ornl.gov/cgi-bin/dsviewer.pl?ds_id=183

Lang et al., (2002):

https://www.aai.ee/bgf/ger2600/

Toolik, (2017):

https://ecosis.org/#result/1d0cb17c-0c0a-4775-8ca6-b8f2975b5041

Dennison and Gardner, (2018):

https://ecosis.org/#result/060d2822-f250-4869-b734-4a92450393f0


**Supporting Information**

The supplementary files of this article can be used to inspect the observed variation in both optical properties and leaf angles by species, and to recalculate the PFT means following different PFT definitions. Four supplementary files are provided: Our recommendation for enhancing CLM5 optical properties table ('S1_CLM5.pdf'), and two source files ('S2_OP.csv' and

'S3_LIA.csv'), which contain species mean optical properties (i.e. reflectance, transmittance and albedo values) values over the VIS and NIR bands, and species mean *LIA*s (in degrees and departure from spherical + classic leaf angle type) along with references to original data. In addition, a pdf copy of the leaf angle tables presented by Ross (1981) ('S4_Ross_1981.pdf') is included as it contains the data used in this study and references to original works.

**Author contribution**

Majasalmi was responsible for the analysis and had a leading role in writing the paper. Bright participated in writing the paper.

**Acknowledgements,**

The research was funded by the Research Council of Norway, grant number 250113/F20. We thank Aarne Hovi, Miina

Rautiainen, Nea Kuusinen, Petr Lukeš, Pauline Stenberg and Jan Pisek for helpful discussions.

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
