# Peer review of "Evaluation of leaf-level optical properties employed in land surface models"

_Geoscientific Model Development, 2019_

## Referee Comment (RC1) · Anonymous Referee #1 · 26 May 2019

The paper targets an important issue of possible systematic errors in surface albedo values used in LSM-s that has a direct influence to the estimates of energy fluxes. In general the manuscript is well written and the applied methods are explained.

1.20 "...we found the optical properties of the visible band (VIS; 400-700 nm) to be appropriate." : How do you estimate this? Is it based on the relative error in reflected/absorbed solar energy or some other criteria?

1.20 "...CLM default and measured estimates were observed," : What is a "measured estimate"?

1.25 "We also found that while the CLM5 PFT-dependent leaf angle definitions were

sufficient..." : Leaf inclination angle is defined as the angle between leaf surface normal and zenith. Do you mean LIA values ?

1.25 "... introduce the concept and application of 'photon recollision probability' (p)." : The p-theory is already introduced in earlier publications. Here the p-theory is applied and proposed also in discussion to be integrated into LCM-s.

Introduction 2.5 "... canopy foliage density (e.g. Leaf Area Index (LAI,m 2 /m 2 ),..." : canopy foliage density (m2/m3) is not leaf area index :: Right bracket is missing.

2.15 "LSMs (e.g. ... (JULES) (Clark et al., 2011)..." : closing bracket is missing. Check also in other places.

3.10 "While measuring LIA of grasses and crops is relatively straightforward and has been conducted since 1960 using inclined point quadrats (Warren Wilson, 1960)..." : With inclined point quadrats the number of contacts is measured (counted). LIA is estimated from that data.

3.20 "...from a leaf or needle in the canopy will interact within..." : "and" seems to be missing.

3.25 "...shoot spectra based on shoot geometry (= p)..." : what is "(= p)"?

M&M 6.5 "For example, dataset by Hovi et al. (2017) contain..." : contains

7.10 "... if spectra were available >2400 nm, it was removed ..." : what was removed?

8.15 "...(and may vary e.g. from 0.12 to 0.28)..." : When STAR is greater than 0.25, then SSA_shoot>SSA_needle according to Eq. (1)!

9.5 "The SSA (shoot) spectra were multiplied with normalized SI spectra for VIS and NIR..."

: Here and in other places: check that lambda is in the subscript where spectrum is pointed to.

[Figure]

10.10 "...and the mean measured estimate ..." : What is measured estimate?

13.Table 3. Please present 95% confidence intervals (or at least standard error) for the mean values.

References

23.10 "Rautiainen, M., Mõttus, M., Yáñez-rausell, L., Homolová, L. and Schaepman, M. E.: Remote Sensing of Environment A note on upscaling coniferous needle spectra to shoot spectral albedo, ,"

:Yáñez-rausell :: albedo, ,

:Please check carefully all records in the list of references, there are many formatting errors (journal names, special characters (si× conifers), latin names) and also typos.
* * *

---

## Referee Comment (RC2) · Anonymous Referee #2 · 6 Jul 2019

Many land surface models require vegetation optical properties (leaf and stem reflectance and transmittance, leaf angle distribution), which are used to calculate the absorption and reflection of solar radiation by vegetation. These parameters are an important part of the model and its surface flux calculation, and are also important in determining changes in surface fluxes related to land cover change (e.g., through changes in surface albedo). The Community Land Model (CLM5) is one such model. Optical parameters in CLM5 trace their heritage to Dorman and Sellers (1989) - some 30 years ago, with minimal changes since then. The authors of the present study compare the CLM5 values with published measurements and show that the model parameters do not match observations in several notable discrepancies. This is a very

nice study that reminds us that model decisions made many years ago (often for expediency) can be forgotten and perpetuated in subsequent model versions. As this paper shows, there is a need to continually recheck models for their fidelity to observations.

Major comments

1. An obvious question is whether the updated optical properties improve CLM simulation of surface albedo, or whether there are other factors in CLM that lessen the influence of the optical biases. Simulations with CLM would be quite helpful in this respect, but are not necessary for publication. What is necessary, however, is a discussion of this issue. My intuition is that it is quite likely the model has been tuned over its many versions to reduce the influence of albedo biases. Or if not explicitly tuned, there are likely compensating errors. A particular example is soil color, which is used to obtain soil albedo. No global soil color dataset was available during model development. Instead, soil color originally came from BATS (1986), which as with Dorman and Sellers is undocumented, but soil color was subsequently estimated by tuning the CLM simulated surface albedo to match MODIS (Lawrence and Chase, 2007; JGR, 112, G01023). In dense canopies with high LAI, this many not be too important but in sparse canopies (LAI < 2) soil albedo becomes more important. Also, CLM blends the optical properties of green leaves and wood (stems) to get effective parameters used in the two-stream radiative transfer (RT) model. This is a huge assumption, and is likely a large source of error. Other problems, as the authors have noted, relate to the simplified plane-parallel, homogenous turbid medium assumption used to model RT and the lack of foliage clumping. The authors have a discussion on page 16 about the need to update model parameters. I would like to see this discussion put in the context of other assumptions and simplifications used in the RT model so that readers can assess for themselves how important the new parameters are for improving CLM.

2. The manuscript evaluates the optical properties of grass and crop leaves, but not stems. This omission must be noted and discussed. The implication of the manuscript is that the updated parameter table is better. Modelers may adopt the new parameters,

**[GMDD](GMDD)**

Interactive
comment

assume they are better, cite this manuscript as the source of the data, but forget that stem optical parameters were not updated for herbaceous plants.

3. The authors mention photon recollision probability (p) in several places throughout the manuscript and make recommendations as to its importance (abstract; introduction; methods; discussion). This is used to upscale from an individual needle to a shoot with many needles – seen in the single scattering albedos (SSA) presented in Table 3 and calculated with eq. (1). The emphasis on p throughout the manuscript, and the recommendation to include it in models, distracts from the manuscript. The simple message of the study is that reflectance, transmittance, and leaf angle used in CLM can, in some cases, differ from measurements and should be improved. This, however, gets conflated with a second message that RT models are using the wrong optical properties and should use shoot values rather than needle values. The author's do not demonstrate that shoot values improve the model compared with needle values. A skeptic is likely to conclude that leaf optical properties in models are wrongly specified, but why fix them because the model should be using shoot values (though this is not proven). It is fine to maintain the distinction between needle and shoot SSA, but the importance of this has not been demonstrated.

Minor comments

page 1, lines 24-26: The authors refer to needle albedo. Is this single scattering albedo (SSA) or reflectance (R)? Presumably it is reflectance (because the comparison is with CLM) but the authors need to clarify because they distinguish between reflectance and SSA in the manuscript.

page 3, lines 8-10: Leaf angle (LIA) is used more fundamentally to obtain the direct beam extinction coefficient, not just to obtain sunlit/shaded leaf area or for RT model inversion.

page 4, line 6: clarify that leaf and shoot albedo refers to single scattering albedo

page 4, lines 20-21: This paragraph is a correct history of how the optical parameters used in CLM were obtained. The sentence "Based on CLM grass and crop ..." is correct, but should be rewritten more strongly by deleting "it seems". Change to: "SiB-table class 7, groundcover, was used ..."

page 5, caption to Table 1: What is meant by "user-friendly version"? Table 1 is the same as in the CLM5 technical description, but collapsed to eliminate equivalent data entries. I believe this is what the authors mean by user-friendly, but that expression is likely to confuse readers.

page 10, line 21: Clarify that 0.07 and 0.05 are from CLM. Compare this sentence with the next sentence, in which the distinction between CLM and observations is clear.

page 11, line 1: Only panel c of Fig. 3 is cited. Panels a, b, and d should be cited when discussing the appropriate PFTs.

page 11, Figure 2: I did not find this figure to be too helpful. There is too much information (too many symbols, too many different PFTs in a panel, both VIS and NIR, both observations and CLM). Perhaps more panels (one for each PFT or for similar PFTs) would be helpful.

page 13, second line from bottom: Change Fig. 2b to Fig. 4b

page 15, lines 2-11: Clarify that this text is for leaves only (not for stems)

---

## Author Comment (AC1) · 19 Jul 2019

**Responses to reviewers**  https://doi.org/10.5194/gmd-2019-59

We thank the two referees for their positive assessments, and suggestions that helped to improve the manuscript. Besides additions and corrections proposed by the referees, we updated two leaf angle analyses due to our discoveries of a significantly larger dataset of measured leaf angles for temperate and boreal species, and of an 'old classic' leaf angle data compilation presented by Ross (1981). To make the leaf angle compilation tables by Ross (1981) available for a wider public, a pdf copy of the tables ('S4_Ross_1981.pdf') is now included as a supplement of this paper.

**Anonymous Referee #1**

The paper targets an important issue of possible systematic errors in surface albedo values used in LSM-s that has a direct influence to the estimates of energy fluxes. In general the manuscript is well written and the applied methods are explained.

1.20 "...we found the optical properties of the visible band (VIS; 400-700 nm) to be appropriate." : How do you estimate this? Is it based on the relative error in reflected/absorbed solar energy or some other criteria?

> \* 1,20: The sentence was modified and now reads: "we found the optical properties of the visible band (VIS; 400-700 nm) to fall within the range of measured values."

1.20...CLM default and measured estimates were observed," : What is a "measured estimate"?

> \* The typo was corrected. Should say measured "values".

1.25 "We also found that while the CLM5 PFT-dependent leaf angle definitions were sufficient..." : Leaf inclination angle is defined as the angle between leaf surface normal and zenith. Do you mean LIA values ?

> \* Yes, should say leaf inclination angle values. The typo was corrected here in in Table 1 figure caption.

1.25 "... introduce the concept and application of 'photon recollision probability' (p)." : The p-theory is already introduced in earlier publications. Here the p-theory is applied and proposed also in discussion to be integrated into LCM-s.

> \* 1,20 Yes, fair point. The erroneous wording was corrected. It now reads: 1:28-31, "In addition, we propose using separate bark reflectance values for conifer and deciduous PFTs, and demonstrate how shoot-level clumping correction can be incorporated into LSMs to mitigate violations of turbid media assumption and Beer's law caused by non-randomness of finite-sized foliage elements."

Introduction 2.5 "... canopy foliage density (e.g. Leaf Area Index (LAI,m 2 /m 2 ),..." : canopy foliage density (m2/m3) is not leaf area index :: Right bracket is missing.

> \* Thanks. Typo corrected and bracket added.

2.15 "LSMs (e.g. ... (JULES) (Clark et al., 2011)..." : closing bracket is missing. Check also in other places.

> \* We added the missing bracket, and checked the text for more missing brackets.

3.10 "While measuring LIA of grasses and crops is relatively straightforward and has been conducted since 1960 using inclined point quadrats (Warren Wilson, 1960)..." : With inclined point quadrats the number of contacts is measured (counted). LIA is estimated from that data.

*       The sentence was revised as: "While measuring LIA of grasses and crops is relatively straightforward and has been conducted since 1960 using inclined point quadrats by measuring the number of vegetation contacts from which the LIA is estimated (Warren Wilson, 1960),..".

3.20 "...from a leaf or needle in the canopy will interact within..." : "and" seems to be missing.

*       This and the following chapter were revised based on reviewer #2 comments.

3.25 "...shoot spectra based on shoot geometry (= p)..." : what is "(= p)"?

*       We removed the typo.

M&M 6.5 "For example, dataset by Hovi et al. (2017) contain..." : contains

*       Typo corrected.

7.10 "... if spectra were available >2400 nm, it was removed ..." : what was removed?

*       Sentence modified. It now reads (8,16): "Note, spectra >2400 nm was removed in effort to harmonize the spectral range of the different data sets."

8.15 "...(and may vary e.g. from 0.12 to 0.28)..." : When STAR is greater than 0.25, then SSA_shoot>SSA_needle according to Eq. (1)!

*       We added as sentence to point out and explain this: 10,5-7: "Note, when STAR is greater than 0.25, then SSAshoot > SSAneedle, which may happen if shoot structure is abnormal (e.g. shoot has very short needles which only cover the upper side of the twig (Thérézien et al., 2007).

9.5 "The SSA (shoot) spectra were multiplied with normalized SI spectra for VIS and NIR...": Here and in other places: check that lambda is in the subscript where spectrum is pointed to.

*       The lambdas were added to appropriate places.

10.10 "...and the mean measured estimate ..." : What is measured estimate?

*       Should say measured 'value', typo was corrected (assumed to refer to 10,30).

13.Table 3. Please present 95% confidence intervals (or at least standard error) for the mean values.

*       Standard errors have now been added inside the parentheses.

References 23.10 "Rautiainen, M., Mõttus, M., Yáñez-rausell, L., Homolová, L. and Schaepman, M. E.: Remote Sensing of Environment A note on upscaling coniferous needle spectra to shoot spectral albedo, ," :Yáñez-rausell :: albedo, , :Please check carefully all records in the list of references, there are many formatting errors (journal names, special characters (si× conifers), latin names) and also typos.

*       Unfortunately, we had problems with reference management software. All references and citations were corrected.

Anonymous Referee #2

Many land surface models require vegetation optical properties (leaf and stem reflectance and transmittance, leaf angle distribution), which are used to calculate the absorption and reflection of solar radiation by vegetation. These parameters are an important part of the model and its surface flux calculation, and are also important in determining changes in surface fluxes related to land cover change (e.g., through changes in surface albedo). The Community Land Model (CLM5) is one such model. Optical parameters in CLM5 trace their heritage to Dorman and Sellers (1989) - some 30 years ago, with minimal changes since then. The authors of the present study compare the CLM5 values with published measurements and show that the model parameters do not match observations in several notable discrepancies. This is a very nice study that reminds us that model decisions made many years ago (often for expediency) can be forgotten and perpetuated in subsequent model versions. As this paper shows, there is a need to continually recheck models for their fidelity to observations.

Major comments 1.

An obvious question is whether the updated optical properties improve CLM simulation of surface albedo, or whether there are other factors in CLM that lessen the influence of the optical biases. Simulations with CLM would be quite helpful in this respect, but are not necessary for publication. What is necessary, however, is a discussion of this issue. My intuition is that it is quite likely the model has been tuned over its many versions to reduce the influence of albedo biases. Or if not explicitly tuned, there are likely compensating errors. A particular example is soil color, which is used to obtain soil albedo. No global soil color dataset was available during model development. Instead, soil color originally came from BATS (1986), which as with Dorman and Sellers is undocumented, but soil color was subsequently estimated by tuning the CLM simulated surface albedo to match MODIS (Lawrence and Chase, 2007; JGR, 112, G01023). In dense canopies with high LAI, this many not be too important but in sparse canopies (LAI < 2) soil albedo becomes more important. Also, CLM blends the optical properties of green leaves and wood (stems) to get effective parameters used in the two-stream radiative transfer (RT) model. This is a huge assumption, and is likely a large source of error. Other problems, as the authors have noted, relate to the simplified plane-parallel, homogenous turbid medium assumption used to model RT and the lack of foliage clumping. The authors have a discussion on page 16 about the need to update model parameters. I would like to see this discussion put in the context of other assumptions and simplifications used in the RT model so that readers can assess for themselves how important the new parameters are for improving CLM.

* We thank the reviewer for the helpful comments and have revised the discussion to reflect these aspects (17-18, 31-14):

"However, whether or not (and if yes, then to what extent) changes in optical properties result in changes to predicted surface albedo requires LSM simulations since LSMs have been tuned to reduce influences of identified biases and possible compensating errors. For example, in the case of CLM, no global soil reflectance dataset was available during model development and soil reflectance data is now based on the tuning of the CLM simulated surface albedo to match MODIS observations (Lawrence and

Chase, 2007). While this may not be too important in dense canopies with high LAI, in sparse canopies (LAI < 2) soil reflectance becomes more important. In addition, we must consider that the major assumptions of 1D RT models themselves are likely to source some error: CLM employs a simplified plane-parallel, two-stream model based on the homogenous turbid medium assumption with isotropic scattering properties. 1D models commonly ignore stems and branches, but the stems are accounted by the CLM RT model: Optical parameters are calculated as a weighted average of leaf and stem areas (i.e. LAI and SAI). This may introduce possible errors (or biases), because i) empirical data and theoretical basis for more accurate definition of SAI is currently lacking (e.g. CLM5 manual, section 29.5.2 (CLM5):" The existing CLM(CN) algorithm sets the minimum SAI at 0.25 to match MODIS observations, but then allows SAI to rise as a function of the LAI lost, meaning than in some places, predicted SAI can reach value of 8 or more. Clearly, greater scientific input on this quantity is badly needed."); and ii) incompatibilities in vegetation structural descriptions in employed RT models (i.e. MODIS LAI is based on 3D RT model whereas CLM employs 1D RT model), which may lead to erroneous assessments of the absorbed, transmitted and reflected fluxes (Pinty et al., 2004; 2006))."

2. The manuscript evaluates the optical properties of grass and crop leaves, but not stems. This omission must be noted and discussed. The implication of the manuscript is that the updated parameter table is better. Modelers may adopt the new parameters, assume they are better, cite this manuscript as the source of the data, but forget that stem optical parameters were not updated for herbaceous plants.

*    This is a fair point. We have added a sentence to the Discussion reminding the reader that optical properties for grass and crop stems are not provided due to the scarcity of measurements (18,14-15): "Noteworthy is that we did not provide updated optical property values for the stems of grasses and crops due to the scarcity of measured spectral data for these plant components."

Regarding the implication of this, we have added text to the Discussion that encourages the reader to reflect on whether such parameters are necessary, especially in light of the lack of SAI information and the LAI-SAI weighting employed in the two-stream model (18,15-18): "However, considering that information on SAI is currently lacking, and that grass and crop stems are ignored by today's RT models employed in vegetation remote sensing applications (e.g. MODIS LAI algorithm (Knyazikhin et al., 1999) and PROSAIL (Jacquemoud et al., 2000)), optical properties of grass and crops stems could also be ignored in CLM RT simulations to correspond better with MODIS LAI."

3. The authors mention photon recollision probability (p) in several places throughout the manuscript and make recommendations as to its importance (abstract; introduction; methods; discussion). This is used to upscale from an individual needle to a shoot with many needles – seen in the single scattering albedos (SSA) presented in Table 3 and calculated with eq. (1). The emphasis on p throughout the manuscript, and the recommendation to include it in models, distracts from the manuscript. The simple message of the study is that reflectance, transmittance, and leaf angle used in CLM can, in some cases, differ from measurements and should be improved. This, however, gets conflated with a second message that RT models are using the wrong optical properties and should use shoot values rather than needle values. The author's do not demonstrate that shoot values improve the model compared with needle values. A skeptic is likely to conclude that leaf optical

properties in models are wrongly specified, but why fix them because the model should be using shoot values (though this is not proven). It is fine to maintain the distinction between needle and shoot SSA, but the importance of this has not been demonstrated.

*        We agree with reviewer #2 that the reasoning and context for including the photon recollision probability ($p$) was lacking in the first version of the article. Some confusion may have been caused by poor wording choice, which was also pointed out by reviewer #1. The key message we wish to deliver is, indeed, that reflectance, transmittance, and leaf angle values used in CLM can, in some cases, differ from measurements and should be updated. However, we cannot overlook advances taken in the fields of vegetation remote sensing and RT modeling since the SiB-table was formulated - Especially, as some of the advancements (e.g. $p$) are intimately linked with optical properties. Due to lacking literature review and missing citations to published papers, the $p$ may have presented itself in our paper as more of a distraction rather than a piece of the puzzle. We remind reviewer #2 that our paper is not just targeting the CLM community, but a broader LSM community, and proper evaluation of optical properties in LSMs requires a more holistic view, which we believe we now provide in the revised version, in a way that does not distract from the suggested improvements to CLM. Specifically, we chose to omit mentioning $p$ in Abstract, incorporate an appropriate literature review for the Introduction, and review in the Discussion some of the recent advancements done related to RT schemes in LSMs and explain why land surface modelers should care about $p$.

This is the suggested framing for Introduction (3-4, 20-20):

[revised manuscript text omitted]

Minor comments

page 1, lines 24-26: The authors refer to needle albedo. Is this single scattering albedo (SSA) or reflectance (R)? Presumably it is reflectance (because the comparison is with CLM) but the authors need to clarify because they distinguish between reflectance and SSA in the manuscript.

\*        The values are for SSA which is now clarified throughout the manuscript.

page 3, lines 8-10: Leaf angle (LIA) is used more fundamentally to obtain the direct beam extinction coefficient, not just to obtain sunlit/shaded leaf area or for RT model inversion.

\*        The sentence was modified and now reads (3,7-8): "LIA is needed to obtain the direct beam extinction coefficient, and e.g. to separate foliage area into sunlit and shaded parts.."

page 4, line 6: clarify that leaf and shoot albedo refers to single scattering albedo

* This clarification is now made.

page 4, lines 20-21: This paragraph is a correct history of how the optical parameters used in CLM were obtained. The sentence "Based on CLM grass and crop ..." is correct, but should be rewritten more strongly by deleting "it seems". Change to: "SiBtable class 7, groundcover, was used ..."

* This sentence was corrected following the reviewer's suggestion.

page 5, caption to Table 1: What is meant by "user-friendly version"? Table 1 is the same as in the CLM5 technical description, but collapsed to eliminate equivalent data entries. I believe this is what the authors mean by user-friendly, but that expression is likely to confuse readers.

* Fair point, we have changed the wording from "User-friendly" to "Collapsed".

page 10, line 21: Clarify that 0.07 and 0.05 are from CLM. Compare this sentence with the next sentence, in which the distinction between CLM and observations is clear.

* We modified this sentence so that the distinction between CLM and observations is clear.

page 11, line 1: Only panel c of Fig. 3 is cited. Panels a, b, and d should be cited when discussing the appropriate PFTs.

* Citations for all panels were added to text.

page 11, Figure 2: I did not find this figure to be too helpful. There is too much information (too many symbols, too many different PFTs in a panel, both VIS and NIR, both observations and CLM). Perhaps more panels (one for each PFT or for similar PFTs) would be helpful.

* We agree that the figure was hard to follow. In effort to simplify, we added a new panel for crops and grasses. Each CLM optical type now has its own panel, with symbols modified, and with their own layout improved.

page 13, second line from bottom: Change Fig. 2b to Fig. 4b

* Corrected.

page 15, lines 2-11: Clarify that this text is for leaves only (not for stems)

* Sentences were modified to point out the text is about leaves.

**References (new additions)**

Chen, J. M., & Black, T. A. Defining leaf area index for non-flat leaves. Plant, Cell Environ. 15(4), 421-429. 1992.

Chianucci, F., Pisek, J., Raabe, K., Marchino, L., Ferrara, C., & Corona, P. A dataset of leaf inclination angles for temperate and boreal broadleaf woody species. Ann.For. Sci. 75(2), 50. 2018.

Ganguly, S., Schull, M. A., Samanta, A., Shabanov, N. V., Milesi, C., Nemani, R. R., ... & Myneni, R. B. (2008). Generating vegetation leaf area index earth system data record from multiple sensors. Part 1: Theory. Remote Sens. Environ 112(12), 4333-4343. 2008a.

Ganguly, S., Samanta, A., Schull, M. A., Shabanov, N. V., Milesi, C., Nemani, R. R., ... & Myneni, R. B. Generating vegetation leaf area index Earth system data record from multiple sensors. Part 2: Implementation, analysis and validation. Remote Sens. Environ 112(12), 4318-4332. 2008b.

Haverd, V., Lovell, J. L., Cuntz, M., Jupp, D. L. B., Newnham, G. J., & Sea, W. The canopy semi-analytic pgap and radiative transfer (canspart) model: Formulation and application. Agric. For. Meteorol. 160, 14-35. 2012.

He, L., Chen, J. M., Pisek, J., Schaaf, C. B., & Strahler, A. H. Global clumping index map derived from the MODIS BRDF product. Remote Sens. Environ. 119, 118-130. 2012.

Jacquemoud, S., Bacour, C., Poilve, H., & Frangi, J. P. Comparison of four radiative transfer models to simulate plant canopies reflectance: Direct and inverse mode. Remote Sens. Environ. 74(3), 471-481. 2000.

Lawrence, P. J., & Chase, T. N. Representing a new MODIS consistent land surface in the Community Land Model (CLM 3.0). J. Geophys. Res. Biogeosciences, 112(G1). 2007.

Lewis, P., & Disney, M. Spectral invariants and scattering across multiple scales from within-leaf to canopy. Remote Sens. Environ. 109(2), 196-206. 2007.

Majasalmi, T., Korhonen, L., Korpela, I., & Vauhkonen, J. Application of 3D triangulations of airborne laser scanning data to estimate boreal forest leaf area index. Int. J. Appl. Earth Obs. 59, 53-62. 2017.

McGrath, M. J., Ryder, J., Pinty, B., Otto, J., Naudts, K., Valade, A., ... & Luyssaert, S. A multi-level canopy radiative transfer scheme for ORCHIDEE (SVN~ r2566), based on a domain-averaged structure factor. Geosci. Model Dev. 2016, 1-22. 2016.

Naudts, K., Ryder, J., McGrath, M. J., Otto, J., Chen, Y., Valade, A., ... & Ghattas, J. A vertically discretised canopy description for ORCHIDEE (SVN r2290) and the modifications to the energy, water and carbon fluxes. Geosci. Model Dev. 8, 2035-2065. 2015.

Oleson, K. W., Lawrence, D. M.,Bonan, G. B., Drewniak, B., Huang, M., Koven, C. D., Levis, S., Li, F., ... & Thornton, P. E. Technical description of version 4.5 of the Community Land Model (CLM), NCAR Tech. Note NCAR/TN-5031STR, 422 pp., Natl. Cent. for Atmos. Res., Boulder, Colo., doi:10.5065/D6RR1W7M. 2013.

Pinty, B., Gobron, N., Widlowski, J. L., Lavergne, T., & Verstraete, M. M. Synergy between 1-D and 3-D radiation transfer models to retrieve vegetation canopy properties from remote sensing data. J. Geophys. Res. Atmospheres, 109(D21). 2004.

Pinty, B., Lavergne, T., Dickinson, R. E., Widlowski, J. L., Gobron, N., & Verstraete, M. M. Simplifying the interaction of land surfaces with radiation for relating remote sensing products to climate models. J. Geophys. Res. Atmospheres, 111(D2). 2006.

Ross, J. The radiation regime and architecture of plant stands. Junk Publishers, The Hague, pp. 391. 1981.

Wang, W., Nemani, R., Hashimoto, H., Ganguly, S., Huang, D., Knyazikhin, Y., ... & Bala, G. An interplay between photons, canopy structure, and recollision probability: a review of the spectral invariants theory of 3D canopy radiative transfer processes. Remote Sens. 10(11), 1805. 2018.

---

## Author Response (AR2)

[https://doi.org/10.5194/gmd-2019-59](https://doi.org/10.5194/gmd-2019-59)

**Topical Editor Decision: Publish subject to minor revisions (review by editor) (26 Jul 2019) by Leena Järvi**

**Comments to the Author:**

Dear Titta,

You have responded to the minor referee comments well but there are a few technical issues that still require your attention. You have added additional supplementary material S4_Ross_1981.pdf which includes copied pages from the book by Ross. In order to have this appendix, you should obtain written consent from the publisher Springer to re-use the material. I also noticed that in the reference list, the publisher for the book should be Springer. Junk Publishers is the Copyright holder. Please correct this.

Here some small issues I found from the manuscript:

P3, L24: Extra bracket after Pinty 2004, 2006. Same applies to P18, L14.

P3, L26: I would remove comma before and correctly

P18, L2: Should be data are

P18, L19: I think lost should not be included to the sentence

Table 3 text: Should read "Standard error is given inside parenthesis"

**Authors response to the Topical Editor:**

Dear Leena,

We thank the Topical Editor for the positive assessment of our revised manuscript. As the book by Ross (1981) is currently available online ([https://link.springer.com/book/10.1007/978-94-009-8647-3](https://link.springer.com/book/10.1007/978-94-009-8647-3)), we decided to remove the 'S4_Ross_1981.pdf' from supplements and citations to S4 from the text.

- The publisher for the book by Ross was changed from Junk Publishers to Springer.
- Extra bracket removed from P3, L24 and P18, L14.
- The comma was removed (P3, L26).
- The 'is' was replaced with 'are' (P18, L2).
- P18, L19: We could not find this typo. However, if the typo refers to p18, L9-L11, that is a direct quotation from CLM5 manual.
- The Table 3 sentence was revised as suggested.

---

## Author Response (AR3)

https://doi.org/10.5194/gmd-2019-59

Topical Editor Decision: Publish subject to technical corrections (07 Aug 2019) by Leena Järvi

Comments to the Author:

Dear Titta,

Thank you for removing the supplement. Many of the data are behind URLs that might change and thus not be permanent. Whenever possible, the used data should have appropriate DOI created e.g. in Zenodo or other platform specific data ID. This would allow the long-term usage and repeatability of the results.

In addition, you have marked in code availability that there is no code available, but you have made your data post-processing with some scripts and these should be made available.

**Authors response to the Topical Editor:**

Dear Leena,

The dataset specific DOIs or IDs, whenever available, were added to section Data availability.

File 'post_processing_2019_gmd59.txt' was added to section Code availability to provide dataset specific instructions and examples of how datasets were post processed. For reader's convenience, the instructions focus on highlighting differences in post processing of different data sets, as essentially, the same processing steps are repeated for different datasets. Based on these instructions readers can quickly notice of how the datasets differ in terms of post processing needs, and can take any dataset and modify the code to fit different input data. No algorithms were developed in this study, as analysis uses existing R-packages prospectr and RLeafAngle.